# Marine outfall discharges contribute to coastal microplastic pollution and the spread of antimicrobial resistance

Raeesa Bhikhoo[1]*, Krisdan Bezuidenhout[2], Lesego Molale-Tom[1], Charlotte Mienie[1], Carlos Bezuidenhout[1]

1 Unit for Environmental Sciences and Management: Microbiology, North-West Univrersity, Potchefstroom, South Africa, 2 South African Research Chair: Cities, Law and Environmental Sustainability (CLES), North-West University, Potchefstroom, South Africa

* raeesabhikhoo1@gmail.com, 26108445@mynwu.ac.za

## Abstract

Microplastics are widespread in marine environments, with significant contributions from land-based wastewater treatment plants (WWTPs). A desktop study was conducted on regulatory framework for marine outfalls in South African coastal cities. The South African regional policy permits proper disposal of wastewater in a marine outfall provided raw wastewater is treated and will not have an adverse effect on the receiving body. The experimental study investigated the capacity of microplastics to serve as reservoirs for multidrug-resistant (MDR) bacteria originating from WWTP effluent. Experiments challenging the regulations were based on microbiology of the effluent that is discharged through an outfall. Microcosms were set up by spiking seawater with WWTP effluent and adding the collected plastic pieces. Scanning electron microscopy (SEM) was used to determine colonization on the microplastics. After 30 days of microcosm exposure, selective media and incubation conditions were used to isolate Enterobacteriaceae. Pure isolates were tested against 16 antibiotics normally used in human clinical settings. In the initial biofilms directly from microplastics from the WWTPs, several genera generally associated with wastewater treatment were isolated. Dominant species isolated and identified were *Citrobacter* sp., *Escherichia* sp., *Enterobacter* sp., *Serratia* sp., *Klebsiella* sp. and *Pseudomonas* sp.. Several isolates were resistant to the last resort of antibiotics, (doripenem and imipenem; 9% to 27%) and some of these isolates were resistant to up to ten of the antibiotics. These findings highlight that clinically relevant Enterobacteriaceae colonize microplastics and survive in biofilms on these microplastics surfaces. Bacterial infections caused by Carbapenem-resistant Enterobacteriaceae have become a global concern in the fight against bacterial infections. Our findings highlight the need for more data to challenge existing marine outfall policies and the outdated notion stating that dilution alone can solve pollution problems.

**Data availability statement:** The nucleotide sequences were submitted to GenBank database (https://www.ncbi.nlm.nih.gov/genbank/about/) under the accession numbers OR178026 – OR178055. The names of the isolates with the allocated accession numbers can be found in S2 in the Supporting Information. The data is easily accessible and is made public.

**Funding:** This work is based on the research supported in part by the National Research Foundation of South Africa (Grant Numbers: UID:118755, UID:115581 and UID:150936). The views expressed are those of the authors and not of the funding agency. The funders had no role in study design, data collection and analysis, decision to publish, or preparation of the manuscript.

**Competing interests:** Raeesa Bhikhoo performed the Microbiology work under the supervision of Cornelius Carlos Bezuidenhout, Charlotte Mienie and Lesego Gertrude Molale-Tom. Krisdan Bezuidenhout contributed the regulatory framework contents. All authors contributed equally to the manuscript and approved the final version submitted. This does not alter our adherence to PLOS ONE policies on sharing data and materials.

**Abbreviations:** AMR, Antimicrobial Resistance; CWDP, Coastal Water Discharge Permit; DFFE, Department of Forestry, Fisheries and the Environment; DNases, Deoxyribonuclease; FTIR, Fourier Transform Infrared; GA,General discharge Authorization; IgA, Immunoglobin A; Intl1, Integron-intergrases; MDR, Multidrug Resistant; MP, Microplastics; NWA, National Water Act; PE, Polyethylene; PP, Polypropylene; PS, Polystyrene; SEM, Scanning Electron Microscopy; WWTP, Wastewater Treatment Plants.

## Introduction

Ocean pollution is escalating, posing the threat of severe consequences to human and ecological health [1–3]. Plastic is one of the major contributors to ocean pollution and is now known to be an emerging anthropogenic contaminant [4–6]. Mass global production and consumption of plastic products lead to a rise in the quantity being swept into the ocean [7,8]. Most of the plastic debris produced is less dense than seawater, thus allowing these to float in the ocean accumulating in subtropical gyres and landing up on the coastlines and beach fronts [9].

Most plastic objects are buoyant, but some materials, like Polyethylene (PE), Polystyrene (PS), and Polypropylene (PP), are denser than seawater and sink to the bottom of the ocean [10,11]. It is estimated that approximately 8.5 million Mt of plastic sinks to the ocean floor, 244 000 Mt float on the ocean, and a portion of plastic gets caught in ocean currents making its way to an ocean gyre [12,13].

The main causes and drivers of plastic pollution in African aquatic ecosystems are related to human population growth, urbanization, and industrialization, leading to an increase in wastewater and pressurizing wastewater treatment plants (WWTPs) [1,14]. Most WWTPs are not designed to fully remove Microplastics (MP) from the wastewater effluent [15].

South Africa has a vast coastline which is 3 400 km long, consisting of four coastal provinces, namely the Northern Cape, Eastern Cape, Western Cape, and Kwa-Zulu-Natal, rich in diversity and maritime activities [1,16,17]. South Africa has been placed eleventh in the world as a possible contributor to plastic marine debris discharging approximately 900 000 million Mt of plastic into the ocean annually [16,18].

Highest densities of MPs occur near populated coastal areas due to point source contamination, particularly from WWTPs, given that these are not designed to fully remove MPs from wastewater effluent, as indicated [1,15]. The most prevalent kind of MPs in South African beaches comprises microfibres, which have been connected to domestic wastewater and sewage sludge disposal facilities [19].

South Africa utilizes a variety of wastewater treatment technologies such as activated sludge, membrane bioreactors, aerobic granular activated sludge, wastewater ponds, bio/trickling filters, rotating biological reactors, and wetlands [20]. The treatment process begins with a preliminary stage, which involves the use of screens to remove larger debris such as paper, plastic, sanitary items, or foreign material that may disturb the process further by clogging or damaging the plant equipment [16,21]. Primary treatment occurs whereby dissolved organic and inorganic constituents, as well as suspended solids, are removed [16]. This is done by the addition of alum or coagulating agents to aid in the separation of the solid and liquid phases in the water [22]. The secondary phase of the wastewater treatment process is the biological removal of dissolved organic matter. Bacterial pathogens are removed in this process by trickling filters, activated sludge, lagoons, extended aeration systems, and anaerobic digesters [22]. The wastewater then undergoes a tertiary phase, whereby disinfection occurs that aids the removal of pathogenic microorganisms. In South Africa, tertiary wastewater treatment is primarily aimed at eliminating pathogens

and nutrients such as phosphorus and nitrogen to mitigate the risk of eutrophication in water bodies. Chlorine is the most common disinfectant used across the country because it effectively inactivates bacteria, viruses, and protozoan cysts by damaging their cellular structures. The treatment processes that target pathogenic organisms also reduce non-pathogenic microorganisms as a secondary outcome of the treatment processes [23,24]. Finally, the water is discharged into a natural waterway [16].

Section 21 of the National Water Act 36 of 1998 (NWA) grants permission for waste or water containing waste to be discharged through a sea outfall [25]. Responsible disposal of wastewater is allowed due to the dilution factor, dispersion of the effluent plume, and decaying of microorganisms, if it undergoes proper treatment [16]. Wastewater treatment plants in South Africa have site specific requirements for the quality of wastewater that can be discharged into the natural environment which are regulated by the Department of Water and Sanitation (DWAF) [26–28].

In addition to these site-specific limits, the DWAF have issued general authorisation in terms of Section 36 of the National Water Act, 1998, Act No. 36 of 1998 providing guidance to wastewater quality, management and requirements [28]. Schedule 3 of the general authorisation permit states that wastewater discharged into a natural waterway should be regulated by Section 21(f) and (h) of the National Water Act describing the quality and management requirements. According to the general authorisation permit, the water entering the waterway should not pose any detrimental effects on the receiving body and must comply with the health and safety of the public in the vicinity of the activity [28].

A marine sewage outfall is a pipeline that releases treated or partially treated wastewater into the ocean. These outfalls typically release effluent underwater, often at significant depths, to facilitate natural dilution and dispersion of pollutants, including pathogens and organic matter. Wastewater undergoes minimal treatment prior to discharge, relying on the ocean's natural processes to further reduce pathogen levels and pollutant concentrations [29,30].

South Africa is reported to have fourteen marine sewage plant outfalls, as reflected in Fig 1 [1], that discharge huge amounts of sewage into the coastal areas daily. These outfalls also include vast amounts of chemical and industrial waste (Fig 1).

In South Africa, wastewater treatment plants (WWTPs) are classified using a comprehensive risk assessment framework. Plants are categorized as either critical risk or high risk based on their Cumulative Risk Rating (CRR), which evaluates factors including design capacity, operational inflow, technical skills, and final effluent quality. Assessment also covers key performance areas such as process control, maintenance practices, and compliance with wastewater quality standards. WWTPs experiencing significant operational challenges, poor effluent quality, or inadequate management practices typically receive critical risk classification, while those with moderate issues are designated as high risk. This system helps identify facilities needing urgent intervention to protect environmental and public health [31,32].

According to the latest Green Drop report, a comprehensive audit of 850 wastewater systems revealed concerning findings: 208 plants were classified as being at critical risk, while 250 were deemed to be at high risk [27]. These results underscore an overall decline in the condition of wastewater systems between 2013 and 2021, suggesting mismanagement and highlighting the urgent need for improvement [20]. Additionally, as the number of contaminants such as plastics has increased, WWTPs that comply with all the standards for efficient wastewater discharge would not sufficiently remove these plastic particles. This leads to a poor critical state and inefficient monitoring of effluent, which causes the release of effluent or waste that is damaging to receiving waters [16].

Sea water contains many persistent organic pollutants, thus plastic debris in the ocean acts as a cleaning agent by absorbing these pollutants onto the surface of the plastics or MPs [33]. The ingestion of MPs may pose detrimental effects on aquatic organisms as several commercial sea species, including mussels, oysters, crabs, sea cucumbers, and fish, have been shown to consume MPs [34].

While microplastics have the capacity to absorb and concentrate persistent organic pollutants (POPs), this process does not eliminate these pollutants from the ocean; instead, it redistributes them within the marine environment. When marine organisms ingest microplastics, the associated pollutants can bioaccumulate and bio-magnify through the food

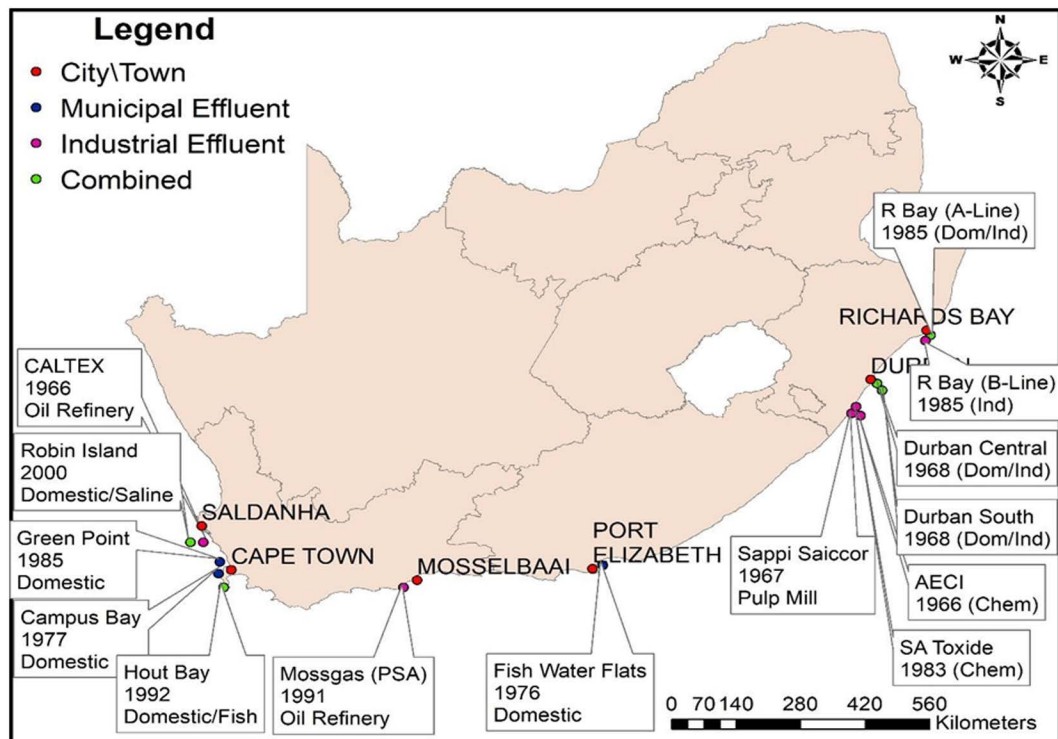

**Fig 1. Effluent discharge sites along the South African coast.** Map adapted by Lohan Bredenhann in ArcGIS Desktop 10.7 using open-source coast-line data (Natural Earth) and outfall locations compiled from the Department of Environmental Affairs' (DEA) 2013–2014 South African National Magisterial Dataset. This work replaces a reference map from Polley, (2015) for compliance with CC BY 4.0 licensing.

chain, posing risks to marine life and human health. Additionally, microplastics themselves act as a significant source of pollution, introducing harmful chemicals into marine ecosystems and causing physical harm through ingestion and entanglement. As highly persistent pollutants, microplastics worsen environmental challenges rather than alleviating them, solidifying their role as a major contributor to marine pollution rather than being a solution [33,35,36].

Microplastics (MPs) are defined as plastic particles measuring less than 5 millimeters (mm) in diameter. This includes both primary microplastics, which are intentionally produced at small sizes, and secondary microplastics, which are formed through the fragmentation and degradation of larger plastic debris. Plastic particles that are under 1 mm to 10 μm are also categorized as microplastics. They are small in size and resemble food particles allowing ingestion by marine organisms [37–39].

Due to the particulate size of MPs, it is made available for ingestion by marine organisms ranging from zooplankton to mammals who mistake it for food [5,33,40–43]. This poses a hazard to marine organisms because, once ingested, plastic fragments may block their digestive tract leading to internal abrasions, impaired locomotion, cause choking, give a false sense of satiation, impaired feeding capacity, and lead to starvation so that death may occur [4,5,41,42].

Every level of the food web, from the primary producers to the higher trophic level of organisms, is affected by this [44]. The build-up of MPs in marine organisms may pose a threat to human health due to the trophic transfer and ingestion of contaminated seafood [5]. Humans may also be exposed to MP consumption by mistakenly swallowing microbeads from toothpaste and commercial beauty products [45]. The detrimental effects of the ingestion of microparticle consumption by humans include chromosomal alteration leading to infertility, obesity, as well as estrogenic mimicking chemicals causing breast cancer [15,34].

Plastic polymers could adsorb organic matter and chemical substances such as heavy metals, pesticides, antibiotics, and xenobiotics onto their surface [46,47]. This is because plastics serve as ecological habitats in the ocean which, in turn, cleans the seawater from persistent organic pollutants [33]. Such niches promote microbial colonization and bio-film formation on the polymers forming a "Plastisphere" [48]. Antibiotic-resistant bacteria end up aggregating on these free-floating polymers as they pose as islands for pathogenic bacteria [44].

This study aimed to isolate and characterize Enterobacteriaceae species from environmental microplastics that origi-nated from a wastewater treatment plant (WWTP) dispersing into the ocean through a marine sewage plant outfall. This was done particularly in the context that current policies allow for partially treated sewage from coastal cities could be discharged into oceans.

## Methods

### Methodology for the policy-regulation literature

The review of the law and policy was desktop-based and analysed primary sources of law, such as legislation relating to the disposal of effluent by municipalities into the offshore marine environment, with specific reference to the attainment of permitting for such disposal under the National Environmental Management: Integrated Coastal Management Act 24 of 2008. The study further drew upon secondary sources of law and gained significant insights from the National Guideline for the Discharge of Effluent From Land-based Sources into the Coastal Environment as the leading governmental policy instrument on marine outfalls.

### Ethical and legal compliance for sampling wastewater and approval of study

Ethical approval was granted for this study by the North-West University, Faculty of Natural and Agricultural Sciences. It was declared a "No-Risk" study with the following ethics number: FNASREC-NWU-00482–21-A9. No specific permits were required for wastewater sampling at the outfall site, as this study did not involve protected areas, endangered spe-cies, or private land access. The sampling was conducted in accordance with South African environmental guidelines for non-invasive marine research. All field personnel completed institutional indemnity forms acknowledging safety protocols and legal compliance. The Municipality's Environmental Management Department was notified of the sampling activities as a courtesy measure, and they have sent out officials to assist.

### Wastewater effluent collection

Plastic samples were obtained from a WWTP at a coastal municipal plant that discharged preliminary treated effluent into the ocean. The wastewater treatment plant is situated away from the coast but discharges wastewater to the ocean through an outlet that is 4.2 km long, has 34 diffusers at a depth of 54–64 m. Sea level rise has an impact on the operating effi-ciency of the pumping needed for this discharge, necessitating the use of more pumping capacity to discharge against an increased head. The wastewater treatment plant is one of the largest plants and are running below the intended capacity. The plant is able to handle greater flows posing an advantage in terms of adaptation to sea level rise [28].

Plastic debris samples were collected from the wastewater influent as well as the wastewater effluent were sampled. Influent and effluent samples were taken by using the dip-sample technique and plastic pieces were then obtained by using mesh sieves in sizes of 20 µm, 100 µm, and 200 µm with a diameter of the sieve being 200 mm each (Filtration group, South Africa).

The selection of 20 µm, 100 µm, and 200 µm mesh sieves for microplastic sampling in wastewater reflects a compro-mise between methodological feasibility and ecological relevance. Smaller sieves (e.g., < 20 µm) are prone to clogging from organic debris and suspended solids, hindering large-scale sampling [49]. These sieve sizes align with wastewater treatment infrastructure, which typically retains particles above 20 µm, reducing the risk of environmental release [50].

While this approach may underrepresent microplastics <20 µm, it ensures operational efficiency and reproducibility across studies, critical for regulatory and comparative research [51].

The samples were collected in two hours at the site in an approximate area of 2 x 4 m in the effluent flow. The plastic pieces obtained were stored in Petri dishes, kept on ice, and transferred back to the laboratory for further analysis. Wastewater effluent (20 L) as well as a seawater sample (20 L) were also taken to set up a simulation of the outfall scenario. Wastewater influent undergoes preliminary treatment before being discharged into the outfall.

The use of 20 L samples for both wastewater effluent and seawater in the outfall simulation aims to ensure adequate capture of microplastic (MP) particles for meaningful analysis. While wastewater effluent typically contains higher MP concentrations than seawater, MPs in both environments are generally sparse and heterogeneously distributed. Using large volume samples (20 L each) enhances the probability of collecting sufficient MPs for quantitative assessment, methodological consistency, and statistical significance. This methodology represents a compromise between practical sample handling and the need to capture MPs across various size ranges (e.g., 20–200 µm) identified in previous wastewater research, although smaller nanoplastics may remain underrepresented [29,52,53].

## Microcosm set-up

Fig 2 illustrates a microcosm set up using plastics obtained from the environment. Seven different microcosms were set up to examine the different dilution factors with controls. The microcosms consisted of actual seawater and plastic pieces obtained from the coastal municipality's wastewater effluent. The plastic pieces collected on site were divided into equal amounts and distributed between the microcosms. These microcosms were left for 30 days at ambient temperature (25°C–27°C). Table 1 shows the composition and set-up of the microcosms used in this study.

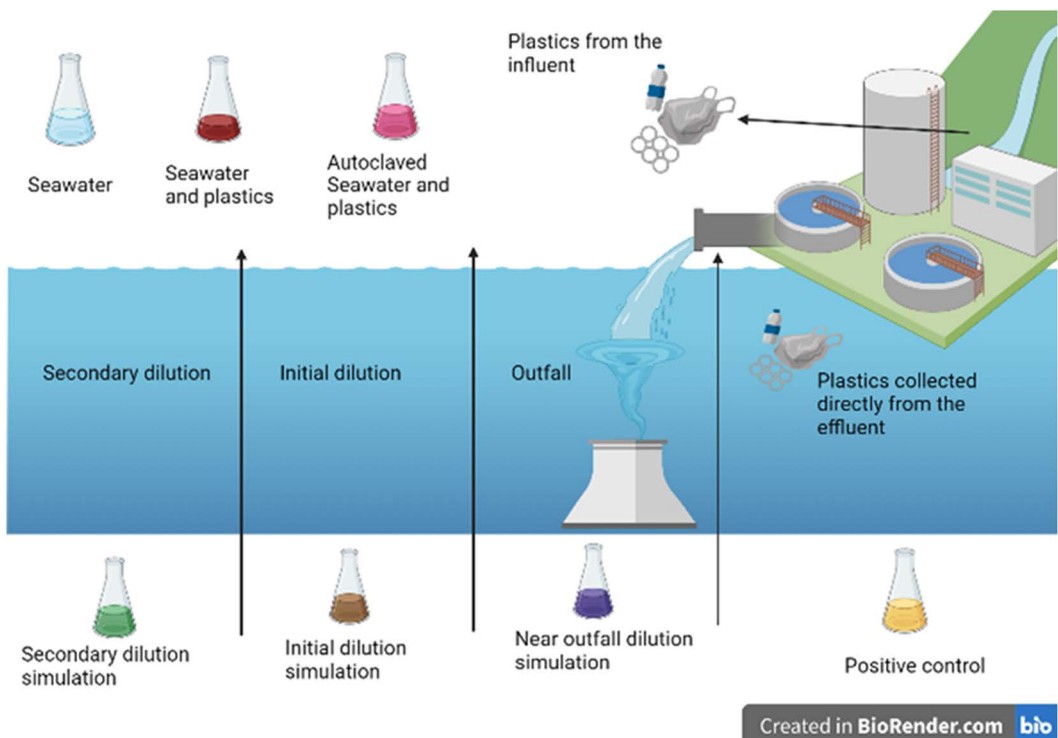

**Fig 2. Schematic overview of the microcosm setup using environmental components.**

**Table 1. Microcosm components and set-up.**

| Microcosm | Description | Dilution Ratio (Seawater: Wastewater) | Volume (mL) | Purpose/ Notes |
|---|---|---|---|---|
| 1 | Simulates ocean outfall near diffuser with physical, chemical, biological, and structural parameters | 3:2 | 150 mL seawater + 100 mL wastewater | Replicates real-world conditions near outfall; assesses environmental impacts and compliance with standards. This dilution might represent a scenario with lower seawater flow or higher wastewater discharge. |
| 2 | Simulates mixing zone at ocean outfall (initial dilution) | 4:1 | 200 mL seawater + 50 mL wastewater | Represents initial dilution zone with plastic pieces from effluent. This dilution could reflect typical conditions where wastewater is significantly diluted by seawater. |
| 3 | Simulates parameter furthest from outfall (secondary dilution) | 9:1 | 225 mL seawater + 25 mL wastewater | Represents secondary dilution zone with highest dilution factor. This dilution might simulate optimal dilution conditions, ensuring minimal environmental impact. |
| 4 | Examines microbial community on plastics exposed to seawater only (furthest dilution) | N/A | 250 mL seawater | Simulates plastics exposed only to seawater without wastewater effluent |
| 5 | Examines microbial community in seawater itself | N/A | 250 mL seawater | Control for microbial community in seawater without plastics |
| 6 | Uses autoclaved seawater with plastics | N/A | 250 mL autoclaved seawater | Control to examine microbial colonization in sterile seawater |
| 7 | Positive control with wastewater effluent and plastics | N/A | 250 mL wastewater effluent | Positive control to assess microbial community with wastewater exposure |

Factors taken into consideration when setting up the microcosms include physical/hydrodynamic factors like flow rates and dilution ratios, chemical/water quality indicators such as salinity, temperature, and pollutant concentrations, biological parameters like bacterial decay rates, pathogen transport, and structural/design factors including port configuration and diffuser length. These parameters are selected to replicate real-world conditions, assess environmental impacts, and ensure compliance with water quality standards. By simulating these factors, a prediction can be made on how wastewater disperses and affects marine ecosystems. The dilution dynamics replicate real-world dilution near outfalls, ensuring compliance with environmental standards [54].

The dilution of wastewater by seawater is a critical parameter. Dilution ratios are determined based on the flow rate of wastewater and the surrounding seawater conditions, ensuring that pollutants are sufficiently diluted to meet environmental standards. The use of these specific ratios as shown in Table 1 is justified because they reflect a range of dilution scenarios that can occur in different oceanic conditions. These ratios allow the assessment of how different dilution levels affect the distribution of pollutants, including microplastics in marine environments.

The microcosms were made up to a volume of 250 mL in sterile Erlenmeyer flasks. The flasks were sealed with sterile cotton wool and covered with foil to eliminate the risk of contamination. All the plastics used in this experiment were rinsed with 1 mL ddH$_2$0 thrice before adding them to the microcosms to eliminate free-living or loosely attached microorganisms on their surfaces.

Rinsing microplastics with ddH$_2$O before adding them to microcosms may remove some loosely attached microorganisms, but it is unlikely to eliminate bacteria embedded in surface crevices or biofilms [55,56]. This approach helps focus on persistent bacterial communities strongly associated with microplastics, as weakly attached cells are more likely to be transient and environmentally variable [57,58]. By standardizing initial conditions, the study can concentrate on long-term microbial colonization, which is critical for understanding the ecological impacts of microplastics in marine environments [59,60].

Plating of each plastic was done on day 30 for the isolation of Enterobacteriaceae and one plastic of each type was kept aside for SEM. Fig 2 is a schematic overview of the microcosm setup using environmental components.

## Visualisation of the plastisphere and the identification of plastics

Scanning Electron Microscopy (SEM) was employed to view biofilms on microplastics as described by [61]. The surfaces of the MPs were viewed with a SEM/ EDS system (Phenom ProX, Thermo Fisher Scientific-US) in high-vacuum mode at 15 kV. Each plastic was analysed, and SEM images captured. Fourier Transform Infrared (FT-IR) spectroscopy was used to characterize the plastics. Samples were preliminary-washed and treated with 1M HCl and then with 30% v/v $H_2O_2$ to remove any loosely associated debris, prior to FT-IR analysis [62]. FT-IR spectra were recorded by using an Agilent Cary 630 infrared spectrometer (Agilent Technologies, USA) equipped with a diamond ATR system with ZnSe crystal. The spectra were taken by using parameters described in [62]. Plastics with matching values > 60% were considered to be valid reads [62].

## Isolation, enumeration, and phenotypical identification of Enterobacteriaceae

Plastics were removed from the microcosm after 30 days and plated onto McConkey agar (Merck, Germany). MacConkey agar is a selective and differential medium used to grow Gram-negative microorganisms based on their ability to ferment lactose (Oxoid Limited, 2021). Enterobacteriaceae was used in this microcosm study to investigate the attachment of the species to plastic pieces derived from wastewater. Enterobacteriaceae are a large family of Gram-negative bacteria, being the focus of this study due to their abundance in wastewater.

The focus on Enterobacteriaceae in wastewater studies, despite the abundance of Pseudomonas, is due to their significant public health and environmental implications. Enterobacteriaceae, including genera like *Escherichia coli* and *Klebsiella pneumoniae*, are common in wastewater and are key indicators of faecal contamination. They are also known for their role as reservoirs of antibiotic-resistant genes (ARGs), making them critical to monitor in wastewater effluents, which can serve as hotspots for the spread of antimicrobial resistance into the environment [52,63]. While *Pseudomonas* species are abundant in wastewater and play a role in biodegradation, they are less representative of faecal pollution and public health risks compared to Enterobacteriaceae. The latter's ability to cause a wide range of infections, coupled with their prevalence in untreated and treated wastewater, highlights their importance in studies focused on microbial risks and antibiotic resistance dissemination. Thus, Enterobacteriaceae are prioritized for monitoring due to their direct relevance to human and environmental health [58,64].

The agar was incubated at 37˚C for 24 hours. Red colonies indicated presumptive *Escherichia coli*, while yellow colonies were presumptive of *Shigella* or *Salmonella*, as well as other coliform bacteria. Clear or colorless colonies were indicative of coliform bacteria. The putative Enterobacteriaceae-isolates were purified by successive streak plating on Tryptone Soy Agar (Merck, Germany). Single colonies were then washed by using the serial dilution method [65] in double distilled water with a dilution factor of 1:10000 or $10^{-4}$. Pure isolates were then streak plated on nutrient agar and subjected to further analysis.

Extracellular enzymes were screened for proteinase activity [66]. This enzyme is identified as a virulence factor as it is responsible for breaking down peptide bonds [67]. Infections caused by pathogens containing proteinase enzymes weakens the human immune system by inactivating the immunoglobin A (IgA) making the host cells more vulnerable to environmental pathogens [41]. Lipase activity was screened using a method described in [68]. This enzyme plays a role in pathogenesis by causing host cell damage and inflammation [69]. DNase activity was tested using a protocol described in [70]. Deoxyribonucleases (DNases) are enzymes that induce the degradation of host DNA. Pathogens then use the degraded DNA as an energy source [71]. Haemolysin activity was tested following a protocol as described in [72]. Haemolysin is one of the first toxins that is used to test for the pathogenicity of the selected colonies by viewing the haemolytic zones that result from the breakdown of red blood cells (Oxoid Limited, 2019). Pathogenicity is the organism's potential to cause disease, and virulence refers to the degree of intensity of pathogenicity [22].

## Antibiotic susceptibility testing

The Kirby-Bauer disk diffusion [73] technique was used to determine the antibiotic susceptibility of selected faecal coliform isolates from the various sites. A homogenous suspension was made by mixing nutrient broth with pure cultures. A 0.5 McFarland standard was used to adjust the densities of the bacterial suspensions. The standard was accepted with the absorbance of an $OD_{600}$ value ranging between 0.08 and 0.1 using a spectrophotometer [74]. Plates were incubated for 24 h at 37°C. Growth inhibition zones were measured in mm after incubation and compared to the NCCLS growth inhibition zone standards for Enterobacteriaceae spp. [75]. The antibiotic disks used in this study can be found in S1 Table within the supporting material. The antibiotics selected for this study was based on the presence of clinically derived AmpC beta-lactamase and extended-spectrum beta-lactamase genes in the environment, particularly in aquatic ecosystems [76,77]. The last resort for the antibiotic treatment of infections caused by Enterobacteriaceae and other Gram-negative bacteria centres on Carbapenems, which are part of the β-lactam antibiotics [78].

## Molecular confirmation of identity

This study incorporated a culture-based technique of isolation which is coupled with molecular confirmation analysis. Genomic DNA was extracted from 1 mL cultures by using a NucleoSpin tissue kit for bacteria (Macherey-Nagel, Germany). The manufacturer's protocol was followed for the extraction of the DNA. The bacterial 16S rRNA, antibiotic-resistant, and virulent gene fragments were amplified by using universal primers and cycling conditions as shown in Table 2.

Three genes were targeted by endpoint PCR: *intI1*, *MOX*, and *FOX*. The *intI1* gene encodes class 1 integron-integrase, which is widely recognized as an indicator of anthropogenic pollution and potential for horizontal gene transfer [79]. *MOX* and *FOX* are AmpC-type β-lactamase genes, conferring resistance to extended-spectrum cephalosporins. These genes were selected due to their known occurrence in wastewater-impacted environments and their association with mobile genetic elements [80,81]. Each reaction tube consisted of a standard 25 µL reaction containing 12.5 µl ready-to-load 2X DreamTaq Master Mix (Fermentas Life Sciences, US) and 0.2 µM of each forward and reverse primer (Applied Biosystem, US) and 50 ng of template DNA. Thermal cycling conditions were as follows: initial denaturation at 95°C for five minutes, 35 cycles of denaturation at 95°C for 30 seconds, annealing for 30 seconds, and extension at 72°C for 60 seconds. The final extension step was executed at 72°C for five minutes [82]. The different annealing temperatures for each PCR reaction are presented in Table 2. The partial 16S rRNA gene of the isolates were subjected to Sanger sequencing as described in [83]. The nucleotide sequences were submitted to GenBank database under the accession numbers OR178026 – OR178055. The names of the isolates with the allocated accession numbers can be found in S2 Table in the supporting information.

## Statistical analysis

MS Excel was used for descriptive statistics in this study to analyze and summarize data effectively. The descriptive statistics likely included measures such as mean, median, standard deviation, and range to provide insights into the distribution and

**Table 2. Oligonucleotide primers used in this study.**

| Primers | Sequence (5′-3′) | Amplicon size (bp) | Annealing temperature | References |
|---------|------------------|---------------------|------------------------|------------|
| 27F | AGAGTTTGATCATGGCTCAG | 1420 | 54°C | [82] |
| 1492R | GGTACCTTGTTACGACTT | | | |
| FOX-F | AACATGGGGTATCAGGGAGATG | 190 | 64°C | [80] |
| FOX-R | CAAAGCGCGTAACCGGATTGG | | | |
| MOX-F | GCTGCTCAAGGAGCACAGGAT | 520 | 59°C | [81] |
| MOX-R | CACATTGACATAGGTGTGGTGC | | | |
| *IntI1*-F | CCTCCCGCACGATGATC | 280 | 54°C | [79] |
| *IntI1*-R | TCCACGCATCGTCAGGC | | | |

variability of microplastic concentrations, sizes, or other parameters in wastewater and seawater samples. These statistical tools help identify trends, compare datasets, and ensure reproducibility in microplastic research by offering a clear numerical summary of the collected data. The cluster formation was drawn up on the basis of antibiotic-resistant data at the different parameters by using Ward's method and Euclidean distances in Statistica 13.3 (TIBCO Software Inc, 2020). Heatmaps were generated by the GraphPad Prism version 8.0.2. The map was adapted by Lohan Bredenhann in ArcGIS Desktop 10.7 and open-access coastline data (Natural Earth). Effluent outfall locations were georeferenced from the Department of Environmental Affairs' (DEA) 2013–2014 South African National Magisterial Dataset. This original work replaces a copyrighted reference map from Polley, (2015) [84] to comply with PLOS licensing requirements. The ArcGIS map was independently adapted by Lohan Bredenhann to avoid copyright restrictions associated with the original reference map. Data sources are cited in Fig 1 caption.

## Results

### Law and policy pertaining to the discharge of municipal effluent into the offshore marine environment

The *National Environmental Management: Integrated Coastal Management Act 24 of 2008* [85] is the specific environmental legislation that deals with the marine environment. Chapter 8 of the ICMA, titled Marine Coastal Pollution Control, deals with the issue of marine outfalls and effluent discharge into coastal waters. The ICMA stipulates that the discharge of effluent originating from a source on land into the marine environment is strictly prohibited except in terms of a general discharge authorisation (GA) or coastal waters discharge permit (CWDP) (Section 69(1)) [85]. The competent authority under the ICMA to issue these authorisations and permits is the Minister of the Department of Forestry, Fisheries, and the Environment (DFFE) (Section 69(2)) [85,86]. The chapter provides specific conditions for those wishing to discharge effluent into the marine environment, which include not wasting water, taking reasonable measures to return freshwater within the effluent to the water source from which it was taken, and compliance with applicable waste standards and water management practices (Section 69(6)) [85].

A coastal municipality wishing to dispose of effluent into the marine environment must comply with the procedure set out for CWDP as contained in the *Coastal Waters Discharge Permit Regulations* [19]. The regulations stipulate the essential details that need to be present in the application. These include comprehensive details regarding the planned discharge facility's scientific, technical, and engineering aspects and processes (Reg 3(2)) [19]. In addition, the applicant must provide insight into the composition of the receiving environment and how the effluent will influence the chemical, physical, geological, hydrological, and biological processes and reactions within the discharge area (Reg 3(3)) [19]. The applicant must also furnish motivations, elucidating why it has not opted for alternative waste management practices (Reg 3(2)(a)) [19].

When deciding CWDP applications, the Minister is guided by the ground rules outlined in the *National Guideline for the Discharge of Effluent From Land-based Sources into the Coastal Environment* [19]. These ground rules serve as a cornerstone of the decision-making process and include considerations such as legislation, management, sensitive areas, environmental quality objectives, waste loads, scientific and engineering assessments and monitoring [19]. The guidelines in 2014:3 differentiate between three coastal zones into which effluent is discharged: surf zones, estuaries and the offshore environment [19]. Each of these zones has a distinct physical process that influences the assimilation of effluent, and the guidelines 2014:24 prescribe minimum treatment measures for these areas [19]. For the disposal of municipal effluent into the offshore environment, the guidelines 2014:33 propose that primary treatment will be required for all new or proposed outfalls, while preliminary treatment may be accepted in cases of existing outfalls [19]. However, these guidelines note that effluent discharges are not the default preferred option in coastal areas, and efforts should be made to prevent and minimize waste disposal into the environment [19].

### Ecological status of the wastewater treatment plant

The wastewater treatment plant that has been sampled has been discharging sewage and selected industrial wastes through two deep-sea submarine outfalls since about 1970. To ensure that the environmental integrity of the region is not compromised, a stringent and comprehensive monitoring programme has been applied. The work encompasses a suite of

microbiological, chemical and ecological measurements that focus on assessing the state of the environment in the vicinities of the two outfall and along adjacent beaches. Additional information is gained through the analysis of effluents and comparative toxicity testing. The treatment plant has 96.3% effluent compliance [28]. Excessive flooding in that area has caused significant damage to the sewage infrastructure resulting in the collapse of access roads, electrical and mechanical failures, and sediment ingress, leading to untreated sewage overflow into beaches, rivers, and the harbour. There has been a significant increase in sewer spills, water leaks and unregulated connections which negatively affect the ecological status of the receiving body [28].

## Visualisation and identification of plastisphere and plastics

Plastics were collected from the influent and effluent wastewater at a coastal WWTP. These were the plastics that would have been directly discharged into the ocean. They were categorized and placed into microcosms to simulate the dispersal potential of biofilm-associated microbes as they disperse into the ocean. The plastics collected were categorized into five different groups, namely fragments, fibre, foam, film, and micro-pellets. The plastics were evenly distributed into the microcosms and incubated for 30 days. Subsequently, they were subjected to SEM imaging and FTIR identification.

The analysis of the SEM images (Figs 3–6) showed that, when a substrate with an appropriate surface area had been introduced into a system, microorganisms formed biofilms that allowed them to attach to the surface of that substrate,

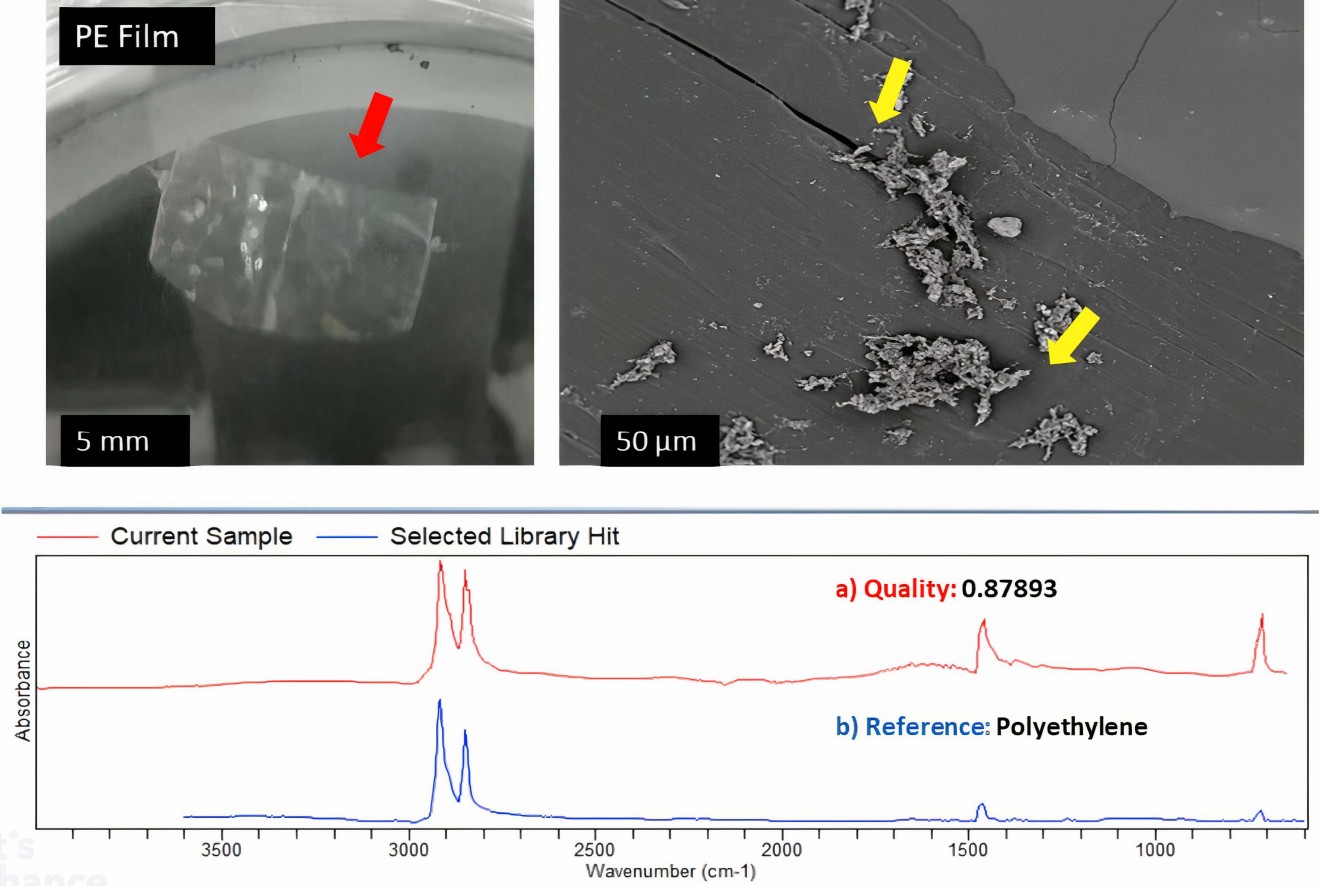

**Fig 3. SEM imaging, FTIR identification, and biofilm formation on the surface of PE.**

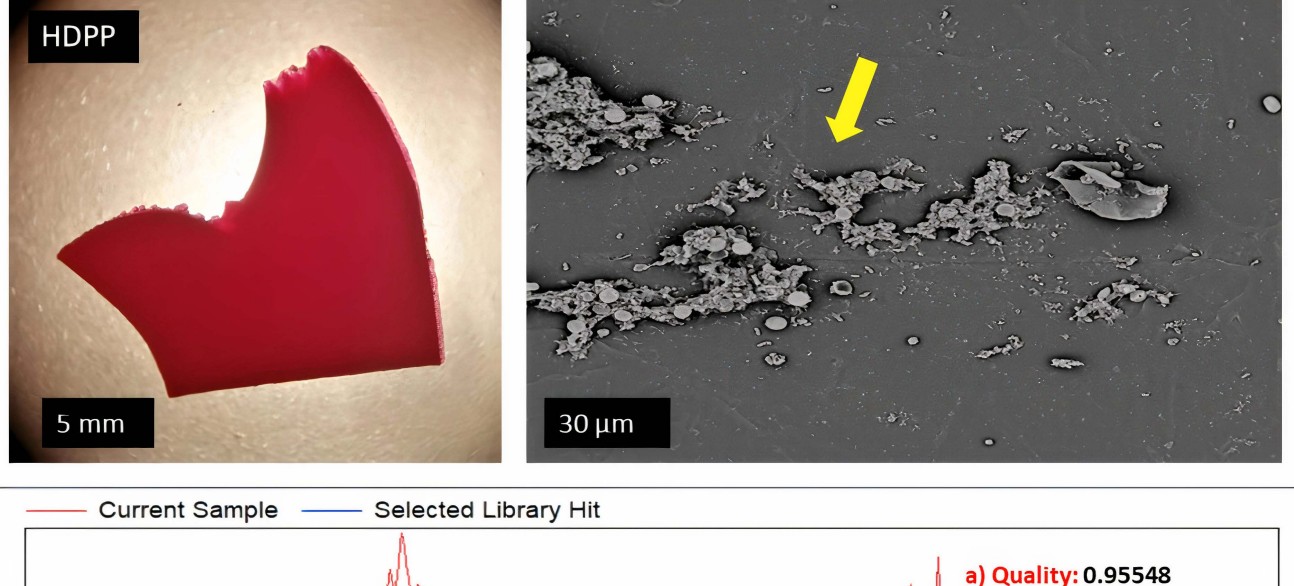

**Fig 4. SEM imaging, FTIR identification, and biofilm formation on the surface of PP.**

increasing their chances of survival and allowing them to disperse widely. Observations based on the SEM images highlighted that the surface of the substrates might influence biofilm attachment, as plastics with a rougher surface with deep and multiple crevices showed the most biofilm formation. FTIR results identified three main types of plastics in the different categories, namely PE, PP, and PS. The most prominent plastic in wastewater identified throughout was PP. This trend is demonstrated in Figs 3–6.

Controls were put into place comparing rough-surfaced polypropylene (PP) and smooth polyethylene (PE) as shown in Fig 7. These controls demonstrates that rough plastics with crevices showed significantly more biofilm formation due to increased surface area and microbial adhesion points, smooth PE substrates still exhibited detectable biofilm colonization, albeit at lower densities. SEM images of PE revealed minimal biofilm clusters, likely influenced by hydrophobicity and chemical properties rather than physical texture alone.

### Isolation, enumeration and phenotypical identification of Enterobacteriaceae

A total of 127 colonies were isolated from the simulations and subsequently enumerated. Of the 127 isolates, 21 were enumerated with a view to the plastics obtained from the influent and a total of 106 isolates from the plastics in the microcosms with a view to wastewater effluent. A larger proportion of the isolates produced proteinase extracellular enzymes. Trends observed in microcosms by using wastewater effluent yielded lower percentages of potential pathogens. Table 3 illustrates the results based on the effluent water source at the various dilution factors.

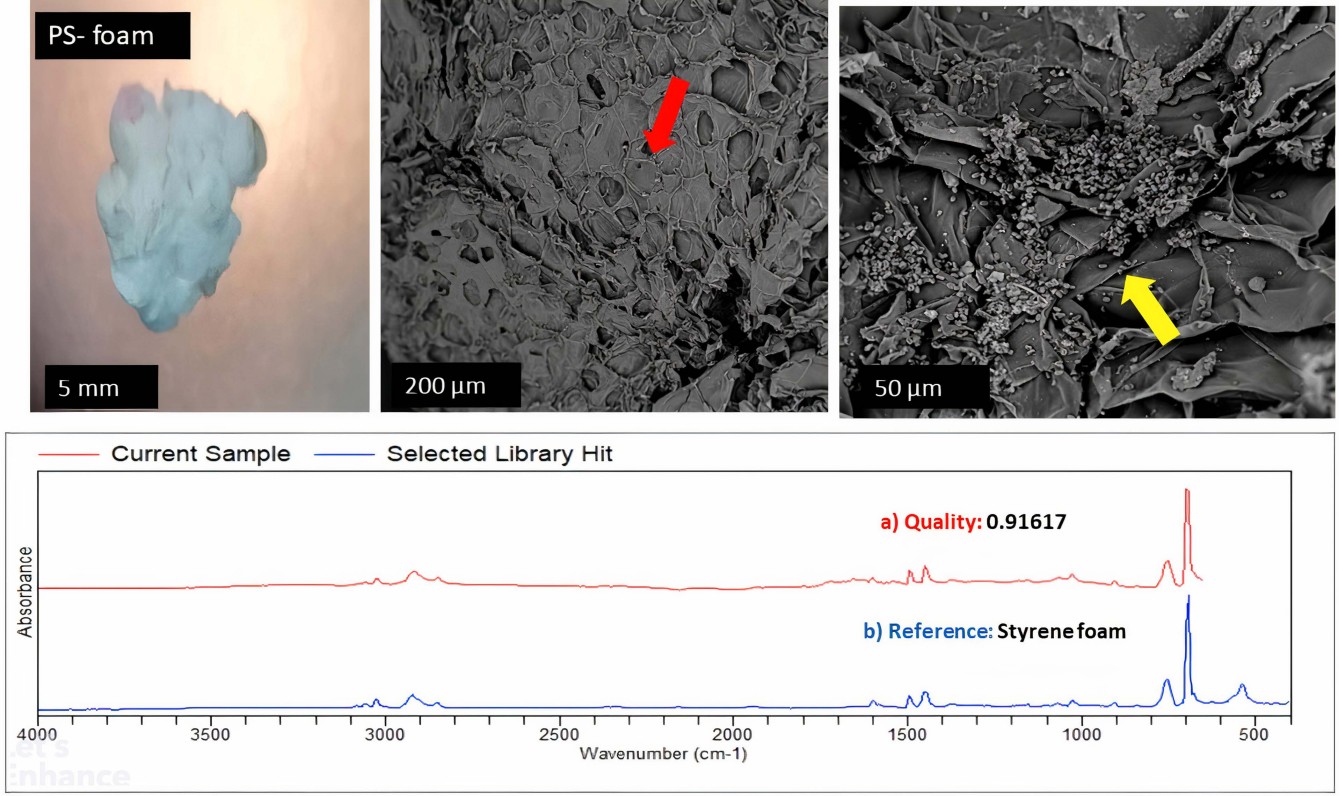

**Fig 5. SEM imaging, FTIR identification, and biofilm formation on the surface of PS.**

## Phylogeny of the isolates on plastics

The 16S RNA Gene for forty-eight (48) isolates were sequenced and illustrated in a phylogenetic tree (Fig 8). Forty-six (46) of these isolates presented haemolysin activity, with two (2) isolates showing no haemolysin activity, but presented Proteinase, Lipase and DNase activity. Fig 8 shows six main clusters of species namely *Pseudomonas, Serratia, Escherichia, Klebsiella, Enterobacter,* and *Citrobacter* spp. The reference strains for the identified species are highlighted in bold. The tree is divided into distinct clusters. *Pseudomonas* isolates shared a cluster with *Stutzerimonas* species. The *Citrobacter, Serratia, Escherichia, Klebsiella,* and *Enterobacter* species had grouped on their own. The bootstrap values were acceptable across the tree, indicating moderate confidence in identities.

The genes tested for by endpoint PCR were *Intl1*, FOX, and MOX. Full circles indicated the presence of the genes (*Intl1* [green], FOX [red], and MOX [blue]). Empty circles indicated the absence of the genes. The MOX gene was present in the genomes of most of the isolates. On the other hand, the FOX gene was present in nine of the 59 isolates screened. The *intl1* gene was mostly present in representatives of *Pseudomonas* sp., *Klebsiella, Citrobacter* sp. and *Enterobacter* spp.. The accession numbers for the isolates that were identified can be found in the Supporting information under S2 Table.

## Antibiotic susceptibility tests

The forty-eight (48) identified potential pathogenic species were subjected to antibiotic susceptibility tests by using the Kirby-Bauer disk diffusion method diluted to the 0.5 McFarland standard. The isolates were tested against 16

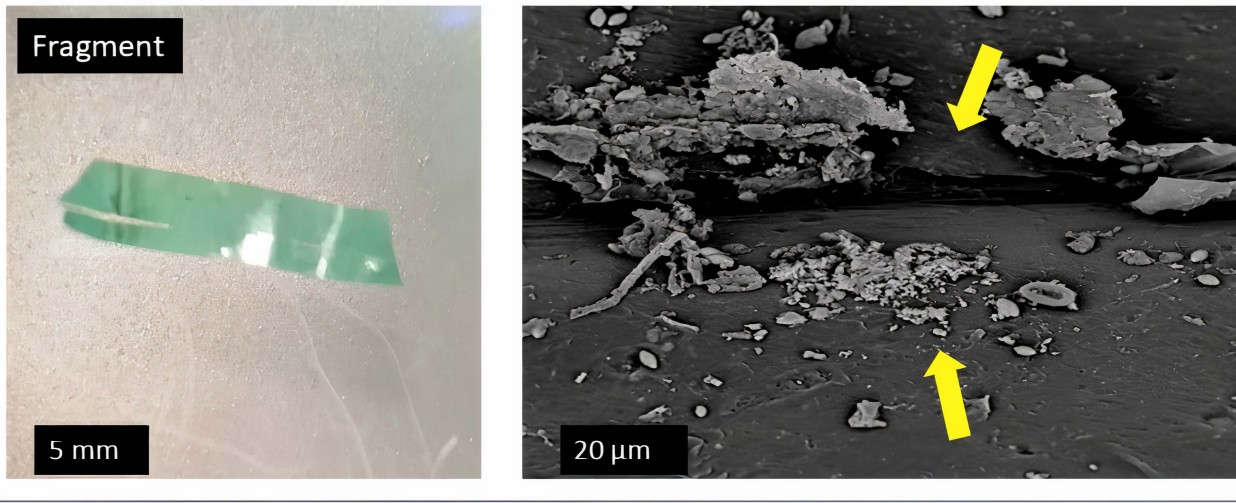

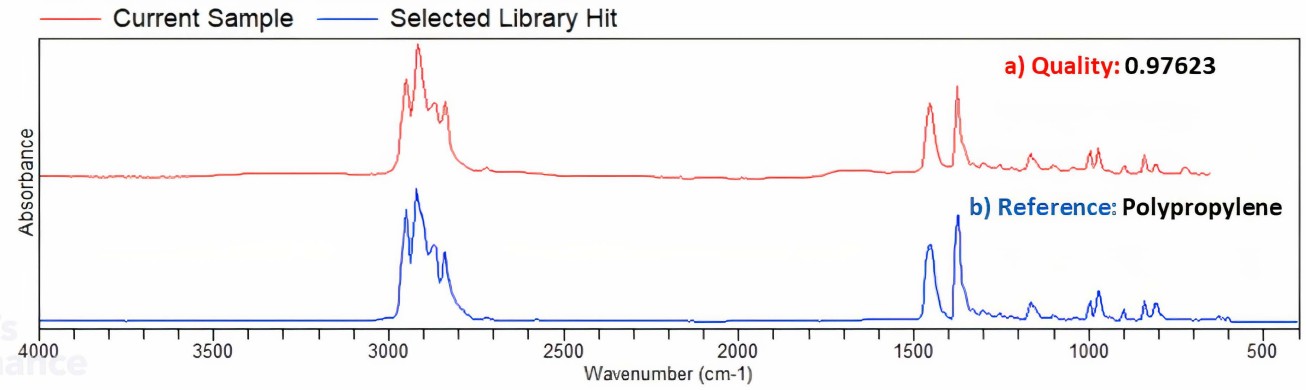

**Fig 6. SEM imaging, FTIR identification, and biofilm formation on the surface of PP.**

different antibiotics belonging to the ten different classes of antibiotics normally tested against Enterobacteriaceae. Fig 9 shows a heatmap with the different isolates showing either resistance (red), intermediate resistance (yellow), or susceptibility (green) with a view to the 16 antibiotics tested. Trends showed a 50% and more isolates were resistant to ampicillin, amoxicillin-clavulanate, trimethoprim-sulfamethoxazole, oxytetracycline, nalidixic acid, and Fosfomycin. Isolates were found to be most susceptible to gentamycin and ciprofloxacin. Isolates were intermediate resistance to Ertapenem and imipenem. This is demonstrating emerging resistance to the carbapenem antibiotics.

## Distribution of different bacterial species on the different plastic types

An RDA ordination plot was used to directly relate environmental and species data. Fig 10 below illustrates an RDA triplot representing the association between the different plastic types sampled in the environment, the Enterobacteriaceae species identified, and their enzyme activities. The statistical association of these parameters was determined by using basic correlation matrices, and marked correlations were significant at $P < 0.05$. One species with specific extracellular enzymes showed a significant positive correlation to one of the three plastics sampled from the environment, as depicted in Fig 10. The correlation and p-value were as follows: (i) Hem_α and PS ($p = 0.042$). There were no statistically significant positive correlations observed for Hem_β, Hem_γ, proteinase, PE, and PP.

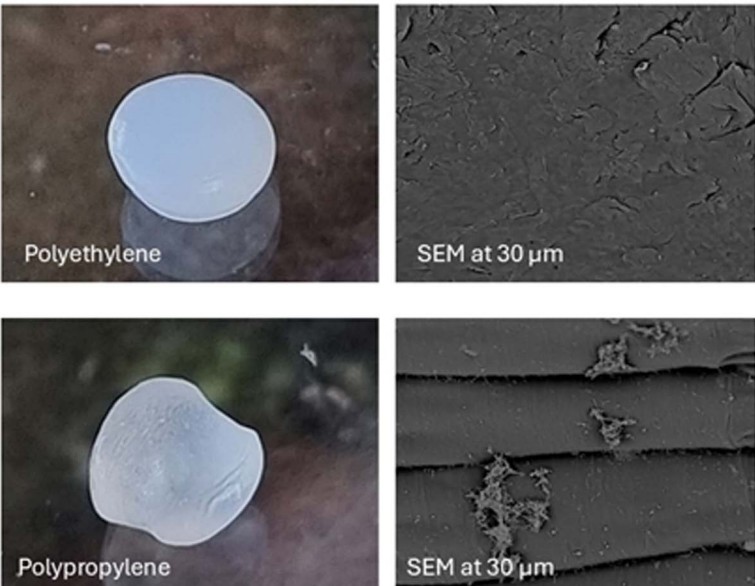

**Fig 7. Comparison of a rough surface plastic (PP) and a smooth surface plastic (PE).**

**Table 3. Enumeration of isolates showing pathogenicity and extracellular enzyme activity from the influent and effluent with dilution factors incorporated into the microcosms.**

| Microcosm and Dilution | Effluent (N = 106) | | | |
|---|---|---|---|---|
| | Haemolysin | Proteinase | Lipase | DNase |
| Microcosm 1 – Near outfall | 6/16 | 6/16 | 0/16 | 2/16 |
| Microcosm 2 – Initial dilution | 5/15 | 9/15 | 0/15 | 0/15 |
| Microcosm 3 – Secondary dilution | 5/14 | 7/14 | 0/14 | 4/14 |
| Microcosm 4 – Furthest dilution | 9/21 | 4/21 | 0/21 | 2/21 |
| Microcosm 6 – Plastics from effluent | 14/20 | 7/20 | 1/20 | 4/20 |
| Microcosm 7 – Positive control | 7/20 | 8/20 | 3/20 | 0/20 |

### Dispersal potential of isolates after exposure from freshwater to salinity

A total of forty-six (46) Enterobacteriaceae isolates from seven different microcosms consisting of wastewater effluent that directly flowed into the ocean through a marine outfall were subjected to cluster analysis. Fig 11 shows the dendrogram assembled by using the antibiotic inhibition zone diameter data to determine the similarities of antibiotic-resistant patterns observed amongst these isolates. The tree is composed of mainly two main clusters, A and B, which are further divided into two sub-clusters each. Sub-cluster A1 is composed further of two minor sub-clusters, A1a and A1b. Sub-cluster A2 is composed of three minor sub-clusters, A2a, A2b and A2c. All these clusters comprised isolates from all the microcosms. Sub-cluster B1 is composed of two minor sub-clusters, B1a and B1b. Sub-cluster B2 comprised minor sub-clusters B2a and B2b. Trends were that clustering occurred mostly among isolates obtained from the plastics at the influent and effluent, but Fig 11 also demonstrates that isolates have the potential to survive and disperse at the furthest dilution with an increased salinity.

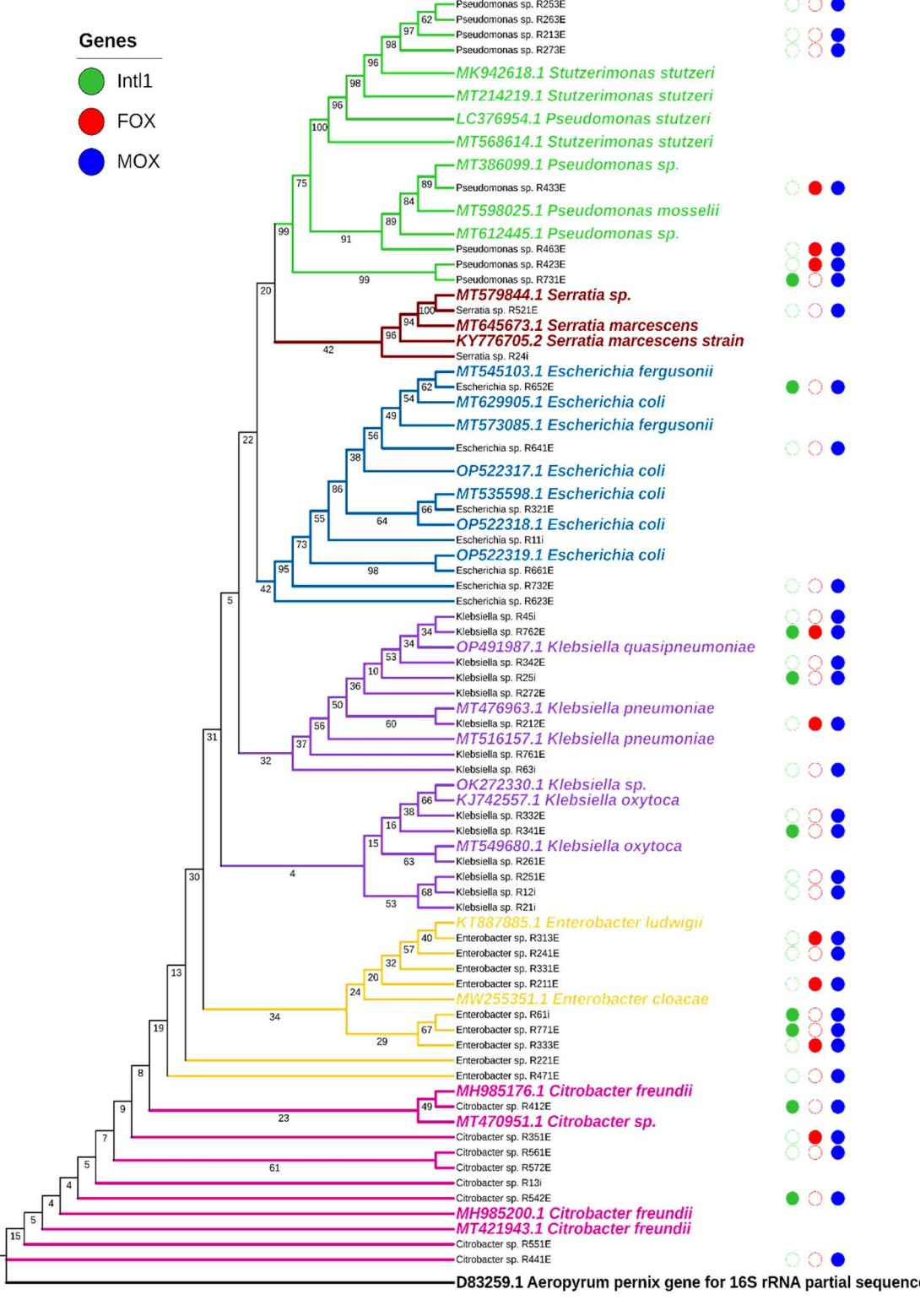

**Fig 8. Neighbour-joining tree showing the phylogenetic relationship of Enterobacteriaceae species.** The reference strains for the identified species are highlighted in bold.

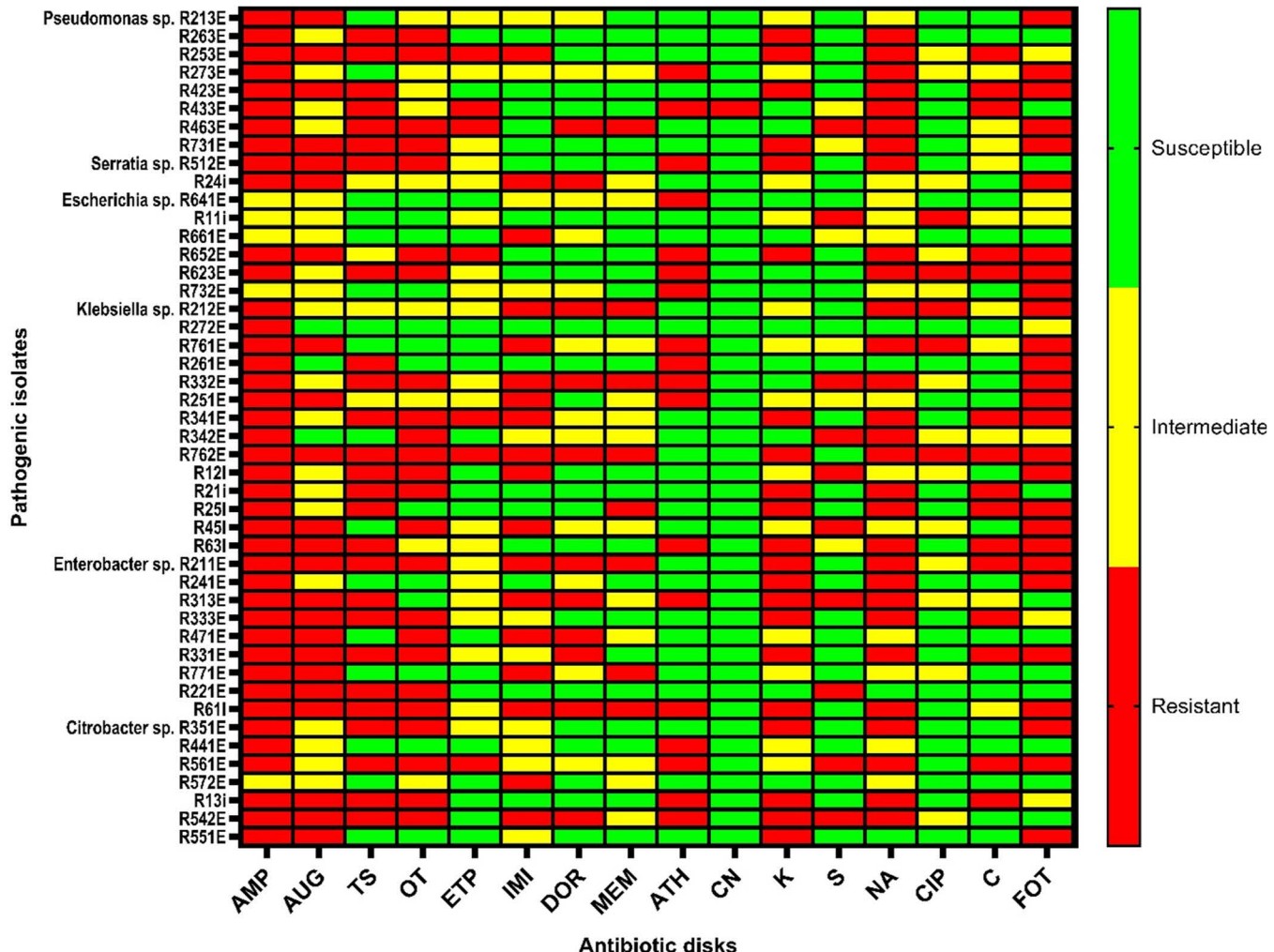

**Fig 9. Heatmap illustrating the antibiotic-resistant phenotype of the pathogenic isolates. AMP** Ampicillin, **AUG** Amoxicillin-clavulanate, **TS** Trimethoprim-sulfamethoxazole, **OT** Oxytetracycline, **ETP** Ertapenem, **IMI** Imipenem, **DOR** Doripenem, **MEM** Meropenem, **ATH** Azithromycin, **CN** Gentamycin, **K** Kanamycin, **S** Streptomycin, **CIP** Ciprofloxacin, **NA** Nalidixic acid, **C** Chloramphenicol, **FOT** Fosfomycin.

## Discussion

This study highlights the dispersal potential of microplastic (MP)-associated Enterobacteriaceae into marine environments, comparing microbial loads on plastics collected directly from wastewater treatment plants (WWTPs) with those subjected to simulated effluent plume dilution. While the National Water Act (Section 21) permits marine outfall disposal of treated wastewater, relying on plume dilution, dispersion, and microbial decay to mitigate risks [16,25], our findings suggest these mechanisms may be insufficient for MP-associated pathogens. Notably, Enterobacteriaceae survived for extended periods (potentially >30 days) when adhered to MPs, enabling far-field dispersal. This aligns with Stark (2016) [87], who reported persistent microorganisms in effluent plumes despite dilution. Our results imply that MP biofilms could protect bacteria from decay, extending their environmental range and challenging current regulatory assumptions about wastewater-derived microbial hazards. The prolonged survival of bacteria released from marine sewage plant outfalls poses significant risks to human health through pathogen exposure and contaminated seafood [7,87,88].

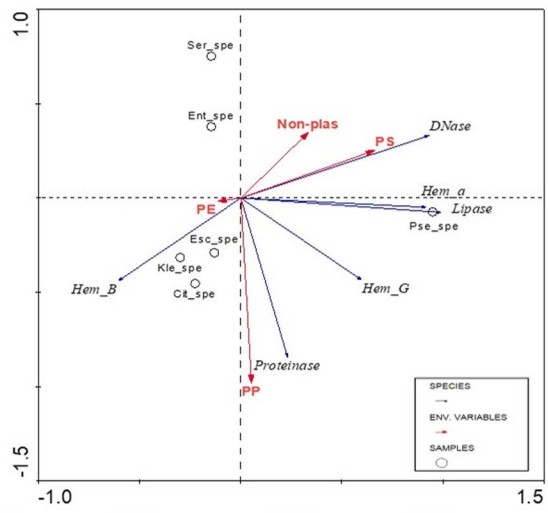

Cit_spe *Citrobacter* species, Pse_spe *Pseudomonas* species, Esc_spe *Escherichia* species, Aer_spe *Aeromonas* species, Ent_spe *Enterobacter* species, Kle_spe *Klebsiella* species, Ser_spe *Serratia* species Hem_a Alpha haemolysis, Hem_B Beta haemolysis, Hem G Gamma Haemolysis.

**Fig 10. RDA plot showing a correlation between plastics and species.**

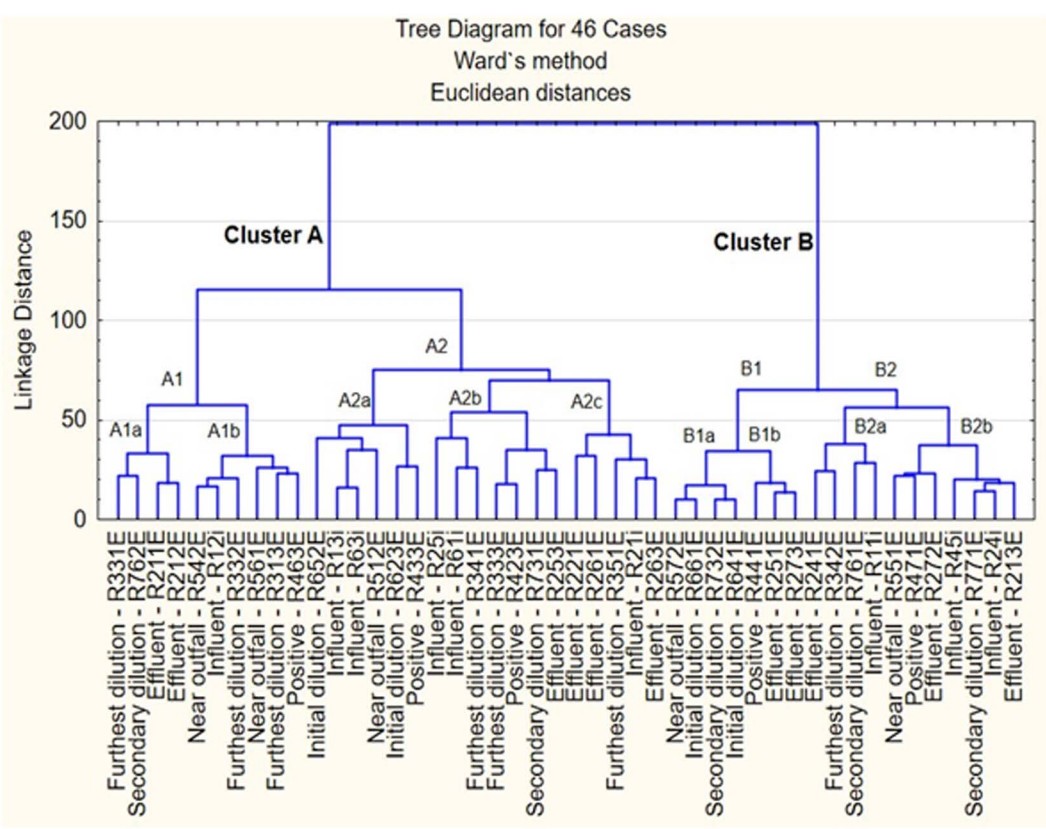

**Fig 11. Hierarchy clustering dendrogram of pathogenic isolates from the influent and effluent.**

The phenotypic and genotypic analyses of plastic-associated isolates from wastewater influent revealed concerning multidrug resistance (MDR) patterns, with all identified species resistant to three or more antibiotics from various classes. As per Magiorakos et al. (2012) [89], MDR is defined as resistance to at least one antibiotic in three or more different antibiotic classes. Particularly notable were two Klebsiella spp. isolates from polypropylene (PP) plastics, which demonstrated resistance to eight of the sixteen antibiotics tested from the different classes. These strains harboured MOX genes, multiple β-lactamase genes, and exhibited virulence potential through extracellular enzyme production. This finding is especially significant as it suggests that microplastics in wastewater systems may serve as vectors for the dispersal of highly resistant pathogens, combining both MDR and virulence factors in a single ecological niche.

The extracellular enzymes (proteinases, DNases, and lipases) produced by these plastic-associated pathogens carry significant ecological and clinical implications. These virulence factors facilitate nutrient acquisition through organic matter degradation [75,90], enhancing bacterial survival in oligotrophic marine environments. More critically, we detected the MOX β-lactamase gene, known to confer resistance to clinically important cephamycins and cephalosporins in hospital isolates [81], in environmental strains. This finding demonstrates a concerning transmission pathway for antimicrobial resistance (AMR), where clinically relevant resistance determinants originating from healthcare settings can disperse into marine ecosystems via wastewater microplastics. The coexistence of these virulence factors and AMR genes in plastic-adherent bacteria suggests MPs may serve as stable platforms for the environmental persistence of high-risk pathogen traits.

Our study identified the class 1 integron-integrase (intI1) gene as highly prevalent among MP-associated Enterobacteriaceae isolates, consistent with its established role as a marker of anthropogenic pollution [91–93]. This finding was expected given its frequent detection in wastewater ecosystems but takes on new significance in the context of microplastic biofilms. The presence of intI1 is particularly concerning as it enables horizontal gene transfer (HGT) of mobile genetic elements, facilitating the acquisition and dissemination of antibiotic resistance genes, virulence factors, and other adaptive traits [92,94]. A common antibiotic-resistant gene associated most with intl1 is blaOXA-10 which encodes an extended-spectrum β-lactamase (ESBL) hydrolysing penicillin and early cephalosporins [91,93]. This association of blaOXA-10 and intI1 is usually found in marine microplastic biofilms [95,96]. When combined with our earlier findings of MDR patterns and virulence factors, these results suggest that MP biofilms may act as hotspots for HGT in aquatic environments, potentially accelerating the evolution and spread of resistant pathogens across clinical and environmental settings.

The discharge of undertreated sewage into aquatic ecosystems remains a critical challenge in low- and middle-income countries (LMICs) [97], with South Africa representing a concerning case study. South Africa's wastewater crisis exacerbates AMR dispersal, with only 23% of WWTPs meeting compliance standards [31]. Non-compliant plants discharge ~12 million liters/day of undertreated effluent [96], while 68% of tested samples exceed *E. coli* limits [98]. This aligns with our findings of intl1 on coastal microplastics suggesting systemic failure to contain AMR. Municipal wastewater management challenges, including systemic governance issues and infrastructural limitations [99], frequently result in the release of non-compliant effluent containing macro- and microplastics colonized by pathogenic biofilms [14,27,100]. This creates an environmental transmission pathway whereby plastic-associated pathogens evade treatment barriers and persist in marine outfall plumes. Compounding this issue, current regulatory frameworks explicitly permit coastal municipalities to discharge partially treated wastewater, creating a paradox where technically legal disposal practices may nonetheless propagate antimicrobial resistance (AMR) and waterborne disease risks.

While plume dispersion models and microbial decay studies suggest rapid diminishment of wastewater contaminants in marine environments with effluent plumes typically becoming undetectable after 72 hours due to lateral diffusion [87,101], these assumptions primarily apply to planktonic microorganisms. Field studies document this pattern, showing marked reductions in culturable Enterobacteriaceae and other indicators following secondary treatment [102] and demonstrating temperature-dependent decay, particularly in tropical waters [103,104]. However, these conventional assessments overlook two critical factors: (1) many bacteria enter a viable but non-culturable (VBNC) state under environmental stress,

evading culture-based detection while maintaining pathogenic potential [105,106], and (2) most importantly, biofilm-associated microorganisms, like those colonizing microplastics in our study, are protected from these dispersion and decay mechanisms. Our findings challenge the prevailing paradigm by demonstrating that microplastic biofilms sustain pathogenic loads far beyond the 72-hour window deemed sufficient for microbial attenuation in current risk assessments.

To accurately model real-world dispersion dynamics, effluent-derived plastics in our study were systematically subjected to six distinct microcosm conditions for 30 days: (1) positive control, (2) autoclaved seawater, (3) near-outfall concentration, (4) initial dilution, (5) secondary dilution, and (6) far-field dilution. This experimental design specifically simulated the progressive dilution gradient of a marine sewage outfall plume. Our findings align with the work of Rozen & Belkin (2005) [103], demonstrating that marine bacteria can not only resuscitate from stress-induced dormancy but also maintain pathogenic potential despite salinity fluctuations. Crucially, our study extends this understanding by showing that microplastic-associated biofilms provide additional protection beyond known resuscitation mechanisms, enabling Enterobacteriaceae to survive dilution stressors that would typically eliminate planktonic counterparts. This microcosm approach reveals how conventional plume dispersion models may significantly underestimate pathogen persistence when accounting for the protective niche of plastic biofilms.

The microcosms have yielded a total of 127 isolates. 21 were enumerated with a view to the plastics obtained from the influent and a total of 106 isolates from the plastics in the microcosms with a view to wastewater effluent. The significant difference in the number of isolates (21 from influent plastics versus 106 from effluent plastics in microcosms) can be attributed to several factors. First, dilution differences and the higher number of effluent microcosms played a role, as effluent conditions simulate wastewater discharge where microplastics interact with larger seawater volumes, enhancing bacterial colonization [84,107]. Wastewater effluent acts as a nutrient-rich hotspot for microbial attachment, promoting biofilm formation on microplastics [108,109]. Additionally, while natural ocean dilution ratios are high, controlled microcosms concentrate microbial activity, intensifying colonization [110,111]. The greater number of effluent microcosms also increases the surface area and environmental variability, raising the likelihood of capturing diverse bacterial isolates [112]. Thus, the observed disparity reflects both experimental design and environmental dynamics. Forty-eight isolates were selected for sequencing because they exhibited pathogenic characteristics, specifically the presence of the hemolysin enzyme, and possessed two or more extracellular enzymes, including proteinase, DNase, and lipase. These enzymes contribute to the isolates' virulence and ability to degrade various substrates, making them of particular interest for further genetic analysis.

Our near-outfall dilution microcosms revealed concerning ecological patterns, with *Citrobacter* sp. and *Serratia* sp. dominating the microbial community and exhibiting multidrug resistance (MDR) to seven or more antibiotics. These isolates carried multiple β-lactamase genes, virulence factors, and the clinically significant MOX gene – a troubling combination of traits for potential pathogens entering marine ecosystems. While the prevalence of *Citrobacter* spp. aligns with expected wastewater coliform profiles [104,113], their MDR burden exceeds typical monitoring assumptions. Notably, *Escherichia* spp. emerged as the sole Enterobacteriaceae with MDR phenotypes in initial dilution conditions, supporting Rozen & Belkin's (2005) findings on *E. coli's* salinity tolerance [103]. However, our study crucially demonstrates that microplastic biofilms appear to expand this survival capacity beyond Enterobacteriaceae's known limits, creating an extended reservoir for antibiotic resistance dissemination in coastal waters.

The genes Intl1, FOX, and MOX were chosen for analysis in wastewater studies due to their critical roles in antibiotic resistance. Intl1 serves as a marker for class 1 integrons, genetic elements that play a key role in the horizontal transfer of antibiotic resistance genes among bacteria. The FOX and MOX genes encode β-lactamase enzymes, which confer resistance to β-lactam antibiotics, a class of drugs essential for treating bacterial infections. By monitoring these genes, researchers gain valuable insights into the dynamics of antibiotic resistance within wastewater systems. This knowledge aids in understanding and mitigating the spread of antibiotic resistance, as well as improving wastewater treatment processes to reduce the dissemination of resistance genes into the environment [52,114,115].

The rationale for selecting these isolates lies in their relevance to antibiotic resistance (AMR) studies. MOX and FOX are β-lactamase genes that confer resistance to β-lactam antibiotics, which are extensively used in clinical and environmental settings [80,116].Screening these isolates helps assess the prevalence and distribution of such resistance genes, particularly in wastewater environments, where antibiotic-resistant bacteria (ARBs) thrive due to selective pressures from pharmaceuticals and horizontal gene transfer [35,117]. By focusing on these genes, the study aims to elucidate their role in microbial resistance dynamics and their potential public health implications, including risks of ARB dissemination into ecosystems [36,63].

Our microcosms simulating secondary and far-field dilutions revealed a persistent consortium of MDR Enterobacteriaceae (*Klebsiella, Enterobacter, Escherichia, Citrobacter*) and Pseudomonadaceae, challenging conventional assumptions about nutrient-limited plume dispersion. Notably, all MDR *Klebsiella* and *Enterobacter* isolates emerged specifically in these diluted conditions, suggesting enhanced survival strategies in oligotrophic environments. This aligns with established ecological theory where nutrient depletion triggers surface attachment [57] and biofilm formation – a phenomenon now amplified by microplastic substrates. Global field studies corroborate this extended dispersal potential, with documented plume transmissions of faecal indicators reaching 200 m in Norway [42], 1.5 km in Australia [87], 1.5 km in South Africa [16], and remarkably 3.8 km in Mexico [118]. Our findings significantly advance this understanding by demonstrating that microplastics not only enable but potentially enhance far-field transport of high-risk MDR pathogens beyond current detection ranges, creating a hidden dissemination network for antimicrobial resistance in marine ecosystems.

The most alarming resistance patterns emerged from microplastic biofilms in autoclaved seawater, where *Pseudomonas* spp. on polypropylene (PP) demonstrated multidrug resistance (MDR) to nine clinically relevant antibiotics – the broadest spectrum observed in our study. These isolates were accompanied by similarly resistant *Klebsiella* and *Enterobacter* strains, creating a high-risk pathogen consortium on plastic surfaces. These findings carry grave implications given the World Health Organization's designation of antimicrobial resistance (AMR) as a top global public health threat [119]. The resistance profiles we observed likely stem from anthropogenic antibiotic misuse – the key driver of resistance gene selection and dissemination [120] – but critically, our study reveals how microplastics may exacerbate this crisis by providing stable environmental reservoirs for MDR pathogens. Unlike transient planktonic bacteria, plastic-adherent biofilms appear to maintain and potentially amplify resistance traits in marine environments, creating persistent hotspots for AMR proliferation even in treated wastewater systems.

These findings underscore the critical need for a One Health approach to address microplastic-mediated AMR dissemination, given the well-documented circulation of resistant bacteria and genes across human, animal, and environmental compartments [120,121]. Our study contributes to growing evidence that microplastics serve as ideal vectors for MDR pathogen spread in aquatic ecosystems [57,92,122,123], with particularly troubling implications given recent pharmaceutical contamination patterns. Liyanage et al. (2023) detected residual amoxicillin and ampicillin – first-line antibiotics for urinary tract infections, respiratory diseases, and bacterial meningitis [124,125] – in 84% of hospital effluents [124]. The coexistence of these antibiotic residues with plastic-borne MDR pathogens creates a perfect storm for resistance selection and maintenance. As our results demonstrate, wastewater microplastics not only transport resistant bacteria but may also concentrate residual antibiotics and resistance genes, effectively creating mobile reservoirs that bypass traditional treatment barriers and threaten all One Health domains simultaneously.

Of grave clinical concern is the emerging carbapenem resistance observed among our isolates, particularly as these β-lactams (ertapenem, doripenem, meropenem, and imipenem) represent last-line antibiotics for ESBL-producing infections [126]. Among 127 tested isolates, we detected concerning resistance patterns to these critical drugs – a troubling finding given their irreplaceable role in modern medicine. Carbapenems serve as essential safeguards against lethal nosocomial infections, especially in immunocompromised patients and complex medical cases [127,128]. The presence of carbapenem-resistant strains in wastewater microplastics suggests these ultimate therapeutic defences are being eroded through environmental pathways, creating a dangerous feedback loop where clinical resistance spreads to the

environment and resistant environmental strains potentially reinfect human populations. This underscores the urgent need to include environmental monitoring of carbapenem resistance as part of comprehensive antimicrobial stewardship programs.

Our controlled microcosm experiments definitively demonstrated that plastic substrates themselves – independent of external environmental factors – sustain pathogenic biofilms, with *Pseudomonas, Enterobacter*, and *Citrobacter* spp. persisting throughout all test conditions. These results align with Metcalf's (2023) findings that plastic-associated biofilms confer significantly longer survival than natural organic substrates [129], fundamentally altering conventional dispersion models. While plume tracking studies typically show microbial attenuation through dilution and decay, our data reveal that microplastics enable pathogenic bacteria to bypass these natural containment mechanisms. The ever-increasing burden of marine plastic pollution creates an expanding network of artificial substrates that facilitate pathogen dispersal far beyond expected plume boundaries [92]. This paradigm-shifting finding suggests microplastics are effectively 'short-circuiting' natural microbial decay processes in marine systems, with potentially grave consequences for coastal ecosystem health and human exposure risks.

The wastewater-derived plastics collected from marine outfall discharge segregated into five distinct morphological categories: films, foams, fibres, fragments, and micro-pellets. FTIR spectroscopic analysis identified polyethylene (PE), polypropylene (PP), and polystyrene (PS) as the dominant polymers, with PP being particularly prevalent in effluent samples. This distribution aligns with established principles of polymer buoyancy and environmental fate - PP's density (0.85–0.94 g/cm$^3$) and durability make it exceptionally mobile in aquatic systems [13,130,131]. These physicochemical properties explain why PP accounts for approximately 24% of marine plastic pollution globally [132], and why our study found it to be the predominant substrate for biofilm formation in wastewater effluent. The predominance of these buoyant polymers creates an extensive dispersal network for attached pathogens, as their low-density characteristics facilitate long-distance transport in surface waters while resisting sedimentation that would otherwise limit microbial spread.

Microplastics (MPs) pervade aquatic ecosystems worldwide due to their density-dependent distribution – polymers either float (PE, PP, PS) or sink (PVC, PET) based on their specific gravity relative to water [130–132]. Our findings corroborate global studies identifying PE, PP, and PS as the most prevalent marine plastics [131], as their buoyancy facilitates long-range transport via surface currents, tides, and wind action [13]. Particularly concerning is PP's dual role as both a highly mobile and biologically reactive substrate: its stable hydrocarbon backbone resists degradation [133] while its naturally rough surface topography promotes extensive biofilm colonization. Artham et al. (2009) demonstrated that PP's micro-crevices and hydrophobic surface create ideal attachment sites for microbial communities [133], explaining our observed predominance of pathogenic biofilms on PP fragments. This combination of environmental persistence and enhanced biofouling potential positions PP as a critical vector for pathogen dispersal – a polymer-specific threat that current pollution mitigation strategies fail to adequately address.

This study demonstrates that polypropylene (PP) polymers serve as preferential substrates for environmental microorganisms due to their characteristically rough surface topography and chemical stability. More critically, we identified PP microplastics as vectors for multidrug-resistant (MDR) pathogens, enabling their long-distance dispersal and prolonged survival in marine environments – challenging conventional assumptions about microbial attenuation in wastewater plumes [42,87,102,118].The convergence of three key factors creates substantial ecological and public health risks: (1) PP's global predominance in marine plastic pollution, (2) its enhanced biofilm-forming capacity compared to other polymers, and (3) the carriage of high-risk MDR pathogens with virulence potential. These findings necessitate urgent revisions to wastewater management strategies and environmental monitoring protocols. Microplastic biofilm screening should be incorporated into outfall monitoring to account for microplastic-mediated pathogen dispersal, particularly as increasing plastic pollution and antibiotic resistance jointly escalate the threats to coastal ecosystems and human health.

## Conclusion

This study demonstrates that treated wastewater effluents can act as reservoirs and vectors for antibiotic-resistant bacteria and genes, especially when associated with microplastic particles. When wastewater is discharged into the marine environment via outfalls, it introduces various pollutants such as plastics, microplastics (MPs), and multidrug-resistant (MDR) bacteria. Bacteria may survive longer and may potentially disperse in the marine environment due to biofilm formation that is used as a survival mechanism. Allowing only preliminary treatment of wastewater under current regulations poses a serious threat to coastal marine environments by facilitating the release and spread of antibiotic-resistant bacteria and their resistance genes.

Microplastics were shown to facilitate the survival and potential dissemination of MDR bacteria in aquatic environments, underscoring their role as environmental contaminants of emerging concern. The detection of clinically relevant resistance genes such as intI1, MOX, and FOX on microplastic biofilms indicates a possible route for gene transfer and enrichment in environmental settings. Most of the isolates form part of the ESKAPE group, which is on the WHO watchlist for emerging pathogens [134]. The ESKAPE group consists of a consortium of bacteria namely: Enterococcus faecium, Staphylococcus aureus, Klebsiella pneumoniae, Acinetobacter baumannii, Pseudomonas aeruginosa, and Enterobacter spp. These pathogens have the potential to "escape" the WWTP and by using plastics as their form of transport. The findings emphasize the need for regulatory frameworks to consider not just chemical pollution, but also the microbial risks associated with microplastics and treated effluents. This study adds critical evidence to the growing concern about the intersection of plastic pollution and antimicrobial resistance, highlighting an urgent need for integrated water quality monitoring and mitigation strategies.

## Supporting information

**S1 Table.** The different antibiotics used in this study with the growth inhibition zone standards for Enterobacteriaceae. (DOCX)

**S2 Table. Accession numbers obtained from GenBank.** (DOCX)

## Acknowledgments

The authors would like to thank:
1. Mr. Willie Landman for the SEM and FTIR work.
2. The DNA sequencing facility of Microbiology.
3. Mr. Abram Mahlatsi, Dr. Kabelo Stenger and Dr. Karabo Tsholo for assistance with sampling.
4. Mr. Lohan Bredenhann for adapting the map.
5. The Aquatic Microbiology Research Group of the NWU, Potchefstroom Campus.

## Author contributions

**Conceptualization:** Raeesa Bhikhoo, Carlos Bezuidenhout.

**Data curation:** Raeesa Bhikhoo.

**Formal analysis:** Raeesa Bhikhoo.

**Funding acquisition:** Carlos Bezuidenhout.

**Investigation:** Raeesa Bhikhoo.

**Methodology:** Raeesa Bhikhoo, Carlos Bezuidenhout, Lesego Molale-Tom, Krisdan Bezuidenhout.

**Project administration:** Raeesa Bhikhoo, Carlos Bezuidenhout, Lesego Molale-Tom, Charlotte Mienie.

**Resources:** Carlos Bezuidenhout, Lesego Molale-Tom.

**Supervision:** Carlos Bezuidenhout, Lesego Molale-Tom, Charlotte Mienie.

**Validation:** Raeesa Bhikhoo, Carlos Bezuidenhout, Lesego Molale-Tom, Charlotte Mienie.

**Visualization:** Raeesa Bhikhoo.

**Writing – original draft:** Raeesa Bhikhoo, Krisdan Bezuidenhout.

**Writing – review & editing:** Carlos Bezuidenhout, Lesego Molale-Tom, Charlotte Mienie.

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
