## [Decision Letter · Decision Letter 0]

2 Mar 2025

Dear Dr. BHIKHOO,

Thank you for submitting your manuscript to PLOS ONE. After careful consideration, we feel that it has merit but does not fully meet PLOS ONE’s publication criteria as it currently stands. Therefore, we invite you to submit a revised version of the manuscript that addresses the points raised during the review process.

We look forward to receiving your revised manuscript.

Kind regards,

Amitava Mukherjee, ME, Ph.D.

Academic Editor

PLOS ONE

Journal Requirements:

3. Thank you for stating the following financial disclosure: This work is based on the research supported in part by the National Research Foundation of South Africa (Grant Numbers: UID:118755, UID:115581 and UID:150936). The views expressed are those of the authors and not of the funding agency. 

4. Thank you for stating the following in the Acknowledgments Section of your manuscript: This work is based on the research supported in part by the National Research Foundation of South Africa (Grant Numbers: UID:118755, UID:115581 and UID:150936). The views expressed are those of the authors and not of the funding agency.We note that you have provided funding information that is not currently declared in your Funding Statement. However, funding information should not appear in the Acknowledgments section or other areas of your manuscript. We will only publish funding information present in the Funding Statement section of the online submission form. 

Please remove any funding-related text from the manuscript and let us know how you would like to update your Funding Statement. Currently, your Funding Statement reads as follows: This work is based on the research supported in part by the National Research Foundation of South Africa (Grant Numbers: UID:118755, UID:115581 and UID:150936). The views expressed are those of the authors and not of the funding agency.

Raeesa Bhikhoo performed the Microbiology work under the supervision of Cornelius Carlos Bezuidenhout, Charlotte Mienie and Lesego Gertrude Molale-Tom. Krisdan Bezuidenhout contributed the regulatory framework contents.

All authors contributed equally to the manuscript and approved the final version submitted.

7. We note that Figure 1 in your submission contain [map/satellite] images which may be copyrighted. All PLOS content is published under the Creative Commons Attribution License (CC BY 4.0), which means that the manuscript, images, and Supporting Information files will be freely available online, and any third party is permitted to access, download, copy, distribute, and use these materials in any way, even commercially, with proper attribution. For these reasons, we cannot publish previously copyrighted maps or satellite images created using proprietary data, such as Google software (Google Maps, Street View, and Earth). For more information, see our copyright guidelines: http://journals.plos.org/plosone/s/licenses-and-copyright.

Reviewers' comments:

Reviewer's Responses to Questions

**Comments to the Author**

1. Is the manuscript technically sound, and do the data support the conclusions?

Reviewer #1: Partly

2. Has the statistical analysis been performed appropriately and rigorously?

Reviewer #1: No

3. Have the authors made all data underlying the findings in their manuscript fully available?

Reviewer #1: Yes

4. Is the manuscript presented in an intelligible fashion and written in standard English?

Reviewer #1: Yes

Reviewer #1: Below mentioned are the query (marked ?) with respect to the text of the manuscript. Clarify Each one separately.

The wastewater then undergoes a tertiary phase, whereby disinfection occurs that aids the removal of pathogenic microorganisms.

How do these pathogenic microorganisms get removed in tertiary treatment, and what about the non-pathogenic microorganisms?

South Africa is reported to have fourteen marine sewage plant outfalls, as reflected in Figure 1

Define marine sewage plant outfalls?

According to the latest Green Drop report, a comprehensive audit of 850 wastewater systems revealed concerning findings: 208 plants were classified as being at critical risk, while 250 were deemed to be at high risk.

On what basis/criteria were these 208 plants classified as being at critical risk, while 250 were deemed to be at high risk?

Sea water contains many persistent organic pollutants, thus plastic debris in the ocean acts as a cleaning agent by absorbing these pollutants onto the surface of the plastics or MPs.

It is a contradictory statement as microplastics are also the cause of pollution.

Due to the particulate size of MPs, it is made available for ingestion by marine organisms ranging from zooplankton to mammals who mistake it for food.

Define the size of MPs?

Influent and effluent samples were taken by using the dip-sample technique and plastic pieces were then obtained by using mesh sieves in sizes of 20 μm, 100 μm, and 200 μm with a diameter of the sieve being 200 mm each (Filtration group, South Africa).

What is the rationale behind taking these sizes as MPs can be smaller than this.

Wastewater effluent (20 L) as well as a seawater sample (20 L) were also taken to set up a simulation of the outfall scenario.

What is the rationale behind doing this simulation?

The microcosms consisted of actual seawater and plastic pieces obtained from the coastal municipality’s wastewater effluent.

What is the amount of Water and plastic taken?

Microcosm 1 was set up to simulate the parameter at the ocean outfall near the diffuser.

What are the parameters?

These simulations were done using the following ratios, 3:2, 4:1, 9:1 of seawater to wastewater respectively and the plastic pieces obtained from the effluent to simulate the parameters in the ocean.

How are these ratios determined, and does it imitate the dilution of wastewater in the ocean?

All the plastics used in this experiment were rinsed with 1 mL ddH20 thrice before adding them to the microcosms to eliminate free-living or loosely attached microorganisms on their surfaces.

Won’t it give the particle results as there are chances of washing off of some the natural fauna of the ocean?

Enterobacteriaceae are a large family of Gram-negative bacteria, being the focus of this study due to their abundance in wastewater.

Why Enterobacteriaceae when literature says Pseudomonas is most abundant in wastewater?

MS Excel was used for descriptive statistics throughout this study.

Explain this. What descriptive statistics were used?

Observations based on the SEM images highlighted that the surface of the substrates might influence biofilm attachment, as plastics with a rougher surface with deep and multiple crevices showed the most biofilm formation.

Have you compared the results with the smooth surface plastic?

Of the 127 isolates, 21 were enumerated with a view to the plastics obtained from the influent and a total of 106 isolates from the plastics in the microcosms with a view to wastewater effluent.

Why is there so much difference in the isolate number? Can it be due to the dilution difference as compared to the oceans?

The 16S RNA Gene for forty-eight (48) isolates were sequenced and illustrated in a phylogenetic tree (Figure 7).

Why only 48 isolates were chosen for sequencing?

The genes tested for by endpoint PCR were Intl1, FOX, and MOX.

Why were these genes taken, mention their importance?

The MOX gene was present in the genomes of most of the isolates., On the other hand, the FOX gene was present in nine of the 59 isolates screened.

Mention the number of isolates in each section clearly, and the rationale behind taking them, as the number of isolates is not constant throughout the study?

Discussion

Completely rewrite the discussion as it is more of results rather than discussion.

**Do you want your identity to be public for this peer review?** For information about this choice, including consent withdrawal, please see our Privacy Policy

Reviewer #1: **Yes: ** Prof (Dr) E. Subudhi

---

## [Author Response · Author response to Decision Letter 1]

5 May 2025

Greetings

Thank you for giving this manuscript a chance. I have addressed all the comments and suggestions as per the decision letter. It can be accessed in detail in the document uploaded "Response to Reviewers".

I am hoping for a great response and a successfully published paper.

Have a great day.

Raeesa Bhikhoo

---

## [Decision Letter · Decision Letter 1]

20 Jun 2025

Addressing the Insufficiency of Marine Outfall Regulations in Mitigating Microplastic and AMR Pollution from Wastewater Treatment

PLOS ONE

Dear Dr. BHIKHOO,

Thank you for submitting your manuscript to PLOS ONE. After careful consideration, we feel that it has merit but does not fully meet PLOS ONE’s publication criteria as it currently stands. Therefore, we invite you to submit a revised version of the manuscript that addresses the points raised during the review process.

We look forward to receiving your revised manuscript.

Kind regards,

Amitava Mukherjee, ME, Ph.D.

Academic Editor

PLOS ONE

Journal Requirements:

Reviewers' comments:

Reviewer's Responses to Questions

**Comments to the Author**

Reviewer #2: All comments have been addressed

2. Is the manuscript technically sound, and do the data support the conclusions?

Reviewer #2: Yes

3. Has the statistical analysis been performed appropriately and rigorously?

Reviewer #2: Yes

4. Have the authors made all data underlying the findings in their manuscript fully available?

Reviewer #2: Yes

5. Is the manuscript presented in an intelligible fashion and written in standard English?

Reviewer #2: Yes

Reviewer #2: (The title does not represent what the work has done.....The short title misguide the reader... I thought the paper would be about regulations and would not have experimental work....

One suggestion for the title is:

“ Dilution of treated sewage in ocean by outfalls negatively impact water quality by harbouring microplastics with multidrug resistant bacteria “ or something like that....

Abstract

I did not understand the last sentence in the abstract:

Did you mean to say dilution of sewage in NOT and answer.....?

Line 30 , in the abstrct you said that : Several isolates were resistant to carbapenems (doripenem and imipenem; 9% to 27%). But you should mention previous authors that have reported that even in secondary treated effluents (many ARB resistant to last resort antibiotics (imipinem, ertapenem, etc) were isolated , as reported by SCIENCE OF THE TOTAL ENVIRONMENT, v. 857, p. 159376-159386, 2023 ) and also from tertiary treated effluent (as Environ Sci Pollut Res 29, 36088–36099 (2022). https://doi.org/10.1007/s11356-022-18749-3).

So, you should highlight the novelty of your work compared to previous studies.

In addition, abstract should contain more results of the study . Half parto f the abstract is explaning regulation about outfall sewage discharge into oceans etc. you should highlight the main results obtained and how they impact and improve the current knowledge about this field (AMR spread from treated domestic wastewater). Such as the following sentence: ....our study crucially demonstrates that microplastic biofilms appear to expand this survival capacity beyond Enterobacteriaceae's known limits, creating an extended reservoir for antibiotic resistance dissemination in coastal waters.

Methods: MIcrocosms set -up

You should show all the different microcosms and characteristics in a Table (with the control and dilution tested and explain better the type of treated effluent (wastewater) from where you collected the microplastics to seed the microcosms.. ; it is hard to follow the rationale and all the details about the microcosms in the text (in results and discussion you explained that, but this should come on Methods. The same is valid for the genes tested : int1, MOX and FOX and why they were chosen to be tested.

How many microplastic particles did you add in each microcosms ? was it fixed and equal in all conditions ? 50 particles ? or there was no number defined / exact ?

Line 504: the 16S rRNA gene....

Line 513: The genes tested for by endpoint PCR were Intl1, FOX, and MOX

Did you presented in methods why you have chosen these genes and the primers used ? what is FOX and MOX stands for ?

Lie 570 to 572: revealed concerning multidrug resistance (MDR) patterns, with all identified species resistant to two or more antibiotics ??? did you mean to sai more than 2 classes of antibiotics ? because, MDR is When the bacteria is resistant to 2 or more than 2 different antibiotic classes, according to Magiorasko. And some bacteria has intrinsic resistance to some antibiotics, did you account for that ? and considered Only acquired resistance ?

Conclusions

Please be more straight foward and say the most importante results , conclusions and impact of the findings of this work. Try to write the 5 most important sentences (as highlights of your study here), and does not repeat results.

**Do you want your identity to be public for this peer review?** For information about this choice, including consent withdrawal, please see our Privacy Policy

Reviewer #2: No

---

## [Author Response · Author response to Decision Letter 2]

15 Jul 2025

We thank the reviewer for all the comments. Revisions have been made. We hope that we are successful this time.

---

## [Editor Report · Decision Letter 2]

21 Jul 2025

Marine Outfall Discharges Contribute to Coastal Microplastic Pollution and the Spread of Antimicrobial Resistance

PONE-D-24-56806R2

Dear Dr. BHIKHOO,

We’re pleased to inform you that your manuscript has been judged scientifically suitable for publication and will be formally accepted for publication once it meets all outstanding technical requirements.

Kind regards,

Amitava Mukherjee, ME, Ph.D.

Academic Editor

PLOS ONE
---

## [Editor Report · Acceptance letter]

PONE-D-24-56806R2

PLOS ONE

Dear Dr. BHIKHOO,

I'm pleased to inform you that your manuscript has been deemed suitable for publication in PLOS ONE. Congratulations! Your manuscript is now being handed over to our production team.

Kind regards,

on behalf of

Professor Dr. Amitava Mukherjee

Academic Editor

PLOS ONE